# "A Picture Is Worth a Thousand Words": Youth Migration Narratives in a Photovoice

Monica Roman [1], Vlad I. Roşca [2,*], Smaranda Cimpoeru [1], Elena-Maria Prada [1] and Ioana Manafi [3]

1   Department of Statistics and Econometrics, The Bucharest University of Economic Studies,
    010374 Bucharest, Romania; monica.roman@csie.ase.ro (M.R.); smaranda.cimpoeru@csie.ase.ro (S.C.);
    elena.prada@csie.ase.ro (E.-M.P.)
2   UNESCO Department of Business Administration, The Bucharest University of Economic Studies,
    010374 Bucharest, Romania
3   Department of Informatics and Economic Cybernetics, The Bucharest University of Economic Studies,
    010374 Bucharest, Romania; ioana.manafi@csie.ase.ro
*   Correspondence: vlad.rosca@fabiz.ase.ro

**Abstract:** This study focuses on the integration facilitators of young migrants in Romania, as resulting from the information gathered through a Photovoice participatory action research method. Young third country nationals were asked to take photos which they thought best summed up their migration experiences. Next to the photos, the migrants were also asked to submit short texts describing the captured images and the meanings that these had for them, thus adding richness and nuance to the data. The evidence gathered reveals that several factors, such as access to education, interacting with a new culture, and with new places and people, are perceived as opportunities and positive migration outcomes. Therefore, human and social capital, as well as the natural or urban environment in the host country, contribute to the enhancement of integration opportunities for young migrants. The paper sets out to analyze how such factors which can facilitate integration are captured in a PAR. Research results show that young migrants perceive their experiences in Romania as enhanced by some factors through which they advance in their integration paths.

**Keywords:** photovoice; participatory action research; migration; youth; migrants' integration



## 1. Introduction

Photovoice, a participatory research and advocacy method combining photography and storytelling to empower individuals [1,2], is increasingly used for understanding migrants' experiences and their perceptions on the new environment at destination [3].Photographs taken by migrants themselves are a means of communication, self-expression, and reflection [4]. Photovoice aims to give marginalized or underrepresented communities a platform to communicate their perspectives, raise awareness about their concerns, and stimulate dialogue with policymakers, researchers, and the broader public [5,6]. In doing so, Photovoice can be a powerful tool for social change, as it engages and amplifies the voices of individuals and communities otherwise overlooked or unheard [7]. The ultimate goal of a photovoice with young, vulnerable migrants would be empowerment [8].Research indicates that Photovoice has a higher capacity than conventional research approaches when it comes to enabling individuals with a refugee background to contemplate positive settlement experiences, rather than solely focusing on the numerous challenges they encounter [9,10].

The aim of this paper is to examine the significant experiences of young vulnerable migrants residing in Romania that contribute to or reflect their integration. The research utilizes photographs as a visual medium to capture the personal journeys made by migrants through integration. The research is an outcome of the large project "*EMpowerment through liquid Integration of Migrant Youth in vulnerable conditions (MIMY)*" (EU Horizon 2020 funding,

grant agreement number 870700), which ran between February 2020 and January 2023. The target group regards young migrants in vulnerable conditions, including refugees or accompanied minors, people in transit, and foreign students or workers, who were generally younger than 30 years old. Therefore, we employ as selection criteria some common conditions associated with the understanding of "vulnerability".

Given that migrants most often arrive in a new country with aspirations and hopes for an improved life, it is crucial to explore the factors that promote their integration and facilitate their adaptation in the host nation.

Migrants often demonstrate resilience and adaptability in navigating a new environment, overcoming challenges, and building a new life [11]. Therefore, they may choose to capture and highlight positive experiences as a way to showcase their ability to thrive in the new community. Photovoice can give them the power to express such experiences and to inspire their peers. Migrants may capture positive experiences that reflect a sense of community, social connections, and belonging. This can include images of cultural celebrations, support networks, community activities, or positive interactions with locals. Personal experiences can affirm a sense of acceptance and integration of migrants within their new environment [12].

Young migrants in particular may document and share positive experiences that reflect their personal aspirations, achievements, and progress. This can include educational pursuits, professional accomplishments, or personal milestones that illustrate their determination and success in their host country. They can even relate to the natural environment, a bridge between two destinations [13]. Migrants may use Photovoice as a means to challenge stereotypes and misconceptions about their communities. They may intentionally capture positive experiences to counterbalance negative narratives often associated with migration, portraying themselves as active contributors to society.

As migrant experiences are multifaceted, both positive and negative aspects—such as language barriers, discrimination, cultural adjustment, or systemic barriers faced by migrants [14]—can be captured and shared through Photovoice.

In our case, the Photovoice PAR method was implemented as part of the MIMY research project. As a collaborative research approach focused on solving real-life problems, PAR involves active engagement of researchers and participants, aiming to empower communities, take action, and create positive social change. In the context of the MIMY project, PAR was employed to actively involve young migrants in reflecting on and sharing their experiences. We are interested in particular in validating the use of Photovoice as a PAR method to explore the facilitators of migrant integration and the factors that ease the adjustment to the new society. Ager and Strang's [15] bi-dimensional model of migrant integration uses structural and cultural factors. Structural integration involves incorporating migrants into the host country's societal structures such as education, employment, housing, and public services. Cultural integration focuses on the social and cultural adaptation of migrants to the host society, including language acquisition, social interaction, and participation in cultural practices and values. Both dimensions are essential for holistic integration outcomes, emphasizing active participation and engagement rather than mere assimilation. As integration is a multifaceted and liquid process, it is expected to encounter various barriers and catalysts simultaneously. In line with this assumption, we use an exploratory approach to identify the most relevant integration facilitators and positive experiences, as transposed by the young migrants in their pictures. Results show that some key facilitators are related to the natural environment, community and people, education, and exploration of a new culture. We noticed that most of the 40 pictures, taken by eight migrants, focused on positive issues. The Section 2 presents the theoretical background, related to results on various highlighted factors that facilitate integration; the Section 3 details the methodological aspects, while Section 4 introduces and analyses the results, highlighting four categories of integration facilitators, before moving on to Section 5.

## 2. Background

Previous research highlighted various integration factors associated with multiple dimensions of integration. For instance, Cimpoeru et al. [11] identified four types of resilience mechanisms used by young migrants: at individual level; family support; formal and informal support received from the community; and support received from the local population. In the same papers, the authors also identified four types of barriers: language barriers and individual level barriers; barriers related to local services and access to education; economic barriers and those related to labor market access; and barriers related to the interaction with the local population. We acknowledge the existence of a large variety of barriers and facilitators [11,16], adjusted to specific integration conditions and groups of migrants. However, an exhaustive review of the entire system of integration factors exceeds the scope of this study. To maintain focus in our discussions and interpretation of results, we have selected papers and topics aligned with the topics identified in our Photovoice. We consider the natural environment, education, people and the community, new cultural experiences, and the architecture as integration catalysts. Overall, our study is integrated within a European dimension of migrant integration, focused on inclusiveness and equal opportunities [17,18].

The accentuation of international migration leads to a complex pattern of *engagement* with the natural spaces inhabited by migrants. For many migrants, engaging with nature is an important way to foster integration and create a sense of belonging in their new environment [19–21]. Lovelock et al. [22] demonstrate that the ability of migrants to locate their own selves in the natural settings of the destination country is key to their integration. Connecting with nature can thus be an important part of helping migrants to integrate, to feel a sense of belonging, and to connect with the local community, while also reducing acculturation stress [23].

The human capital of migrants has a strong impact both on the probability of employment and on wages, being a key trigger to socio-economic integration [24]. Dustmann et al. [25] grouped some of the European host countries in three classes, considering the educational background of migrants: the first group included the United Kingdom and Ireland, characterized by a high-skilled foreign-born population; the second cluster included France, Germany, and the Netherlands, having mainly a low-skilled foreign-born population; the third group included Spain and Italy, where foreign and native populations have rather comparable educational levels.

Bynner and Hammond [26] found that education has more benefits, such as self-confidence, well-being, and self-efficacy. Starting from the hypothesis that migrants are slightly less educated than the native population, De Paola and Brunello [24] explore how the educational policies in the host countries impact the educational outcome of the migrants.

Migrant workers are a vulnerable group of workers, often engaged in what are known as 3-D jobs—dirty, dangerous, and demanding (sometimes degrading or demeaning). Usually, migrants are paid less, work for longer hours, and in worse conditions than native employees, being often subject to human rights violations, abuse, human trafficking, and violence [27]. Migrant workers have higher rates of negative occupational exposures, which mean poor health outcomes, workplace injuries, and occupational fatalities [28]. There are cases in which the migration process includes transit phases from the origin country to a safe destination, and sometimes transit points can turn into final destination. Transit is often dangerous with exposure to smugglers, traffickers, etc. [29]. Many of the health risks for migrant workers come from jobs that are physically demanding or implyenvironmental hazards inherent in the occupational setting, i.e., jobs that involve increased exposure to environmental toxins, extreme temperatures, pesticides and chemicals [27]. Workers with lower education levels and/or with limited language skills tend to incur more occupational injuries than those with higher education levels [30].

Migration can also lead to architectural and cultural changes in urban areas. Migrants can be one group of people particularly vulnerable in urban areas. Socio-economic and spatial clustering may be maintained by constructing ghettoes or ethnic enclaves, but if integration is the objective, then desegregation is the key [31]. If segregation is maintained, different standards of public services could be imposed [32]. It is the role of architecture to create cultural landscapes that promote variety through inclusive designs. Churchman and Mitrani [33] found that migrants have strong allegiances for the characteristics of their native streetscapes. Different researchers found that the countryside is perceived as a safe place [34,35]. Migrants retain and amplify aspects of their home culture in the host culture. When bringing along their own cultural traditions and architectural styles, the blending of the latter leads to diverse and eclectic cityscapes. Moreover, architecture and urban cultures are shaped by continuous waves of migration over the years. Therefore, Rishbeth [36] stressed that landscape architects need to be aware that different ethnic groups may not only have different perceptions from a city, but also various expectations or needs within those. Inclusive architectural designs are thus required. As migration can have an impact on the social and political structure of a city (with new populations bringing new ideas and perspectives), more inclusive urban societies will be required. Social and economic inequalities that migration brings along into cities will need to be addressed. Such urban social inclusiveness might be obtained through sustainable migration, smart participation, and citizenship [37].

To conclude, substantial evidence emphasizes the crucial role of some integration factors such as human and social capital. The literature on cultural, environmental, or architectural capital—although not as extended as that on human and social capital—also presents various interaction types of migrants with their receiving societies. This paper aims to bring a fresh perspective on such factors by observing them through the magnifying lens of a Photovoice.

## 3. Research Design and Data Collection

This study uses Participatory Action Research (PAR) to investigate integration processes of migrants. This novel research method focuses on creating an immersive artistic experience to gain insights about the integration opportunities of migrants and about their transformational journeys [38]. The goal is to empower migrants to critically reflect on their own integration process by providing them with an art-based platform to express ideas, celebrate (home or host) culture, and share their experiences. Another purpose of art-based events is to inspire migrants to directly contribute to the creation of new information and understanding, with little interference from the researchers. Migrants possess invaluable knowledge of their backgrounds and cultures which can be used to create something unique and remarkable. It is this type of knowledge that PAR methods attempt to reach. By making use of resources available to them, such as their communities and modern technology (i.e., a photo camera or smartphone camera), migrants are given the opportunity to create something meaningful (for example an artwork), not only for themselves, but also for the research team. New understanding is thus generated which is free of scholarly biases.

Young migrants aged between 18 and 32 participated in this study. To ensure the heterogeneity of the sample for a more comprehensive study, participants were recruited from two population groups with different origins: migrants from the neighboring Republic of Moldova and migrants from Arabic-speaking countries. These two ethnic groups hold the largest migrant populations in Romania [39]. In line with Tsang [40], we followed several stages for data collection. The participants were recruited through the connections established during the MIMY research project, from the communities of students, and via snowball sampling. In a preliminary stage (July–10 September 2022), the art-based session

was prepared through the exchange of emails and phone calls with potential participants, meant to inform them about the Photovoice activity and to provide necessary instructions.

Recruiting participants was an important challenge, as less than half of the contacted people responded to the invitation and participated in the sessions. The final group consists of eight migrants (all female) from the Republic of Moldova (aged 20 and 22, both students), Lebanon (22 and 25, both employees), Palestine (23, master student), Algeria (29, employee), and Syria (25 and 32, both employees). We noticed that participants adopted a self-selection approach based on their familiarity with arts and visual performance. Of the eight migrant participants in the final event, one is a junior architect, another studies film direction, while the rest are passionate photographers or simply practice it as a hobby. Besides their interest in sharing their personal migration experiences, an interest in visual arts seems to have played a role in their engagement in the Photovoice.

Two sessions were organized, considering the typical arrangements for Photovoice activities. Both sessions were organized in a hybrid mode (on the premises of the Bucharest University of Economic Studies and online), depending on the participants' availability to come in person and in order to maximize the number of people participating in the sessions. The first session, held in mid-September, was designed to introduce participants to ethical and privacy concerns. At the same meeting, the project was presented, instructions were offered, a timeframe was put forward, and the objectives of the activity were defined. A total of 17 participants from Bucharest and Iasi attended the first workshop and were given information via a PowerPoint presentation. Participants were asked to take two to five photos that they considered best captured their migration experiences to or within Romania. At least one of the images had to be accompanied by a brief written description of a maximum of 100 words. Since the research team opted to offer participants enough flexibility to communicate whatever they believed best captured their experiences, no additional cues were provided. Participants were merely instructed to take any kind of photo they felt best conveyed their experience, or to use any of their previously shot pictures that they felt had a particularly powerful message. Also, the participants were given the option to decide whether they wanted to appear in the images (i.e., selfies, portraits) or not. Landscapes and any other types of pictures were welcomed. After the first meeting, the participants were given two weeks to prepare their images and accompanying descriptions.

The second meeting took place in hybrid format. To accommodate the hybrid format, the photos were compiled into a PowerPoint document with accompanying text descriptions, which was projected via beamer in the meeting room. Each participant presented their photos and discussed their meanings in the context of the migration experiences. The session continued with group discussions and reflections on the ideas that emerged. Save for a few who reported difficulties in providing adequate descriptions for their photos, participants generally found ways to follow the requirements. The group discussions in the second session also enabled a collaborative development of the narratives. The photographic activity proved to be a positive introspective exercise, while participants found common ground among themselves.

In October 2022, a vernissage was held containing a selection of the most relevant pictures from the Photovoice. The pictures were exhibited as large posters and medium-sized framed photos placed on tables, in a library room of the Bucharest University of Economic Studies. During the two-hour presentation, participants gathered around the posters, engaged in free group discussions, and shared their Photovoice activity experience with other stakeholders, researchers, and members of NGOs and associations. The researchers organized the photos submitted by participants and paired them with the corresponding narratives; a printed photo album that serves as a tangible memento of the event and as lasting legacy was also produced.

To gather information for this research paper, we used an ethnographic analysis and examined a variety of sources, including photos with their descriptive accompanying texts and hand notes of the research team from the three participatory sessions. Our

interpretations are concerned with migration and integration factors and were based on anthropologically founded theoretical frameworks. Our further results rely on both photos and associated narratives. As already mentioned, several topics emerged from our analysis, related to nature, education, community and people, places, and architecture. A tendency among participants to represent more positive experiences rather than negative ones was observed.

## 4. Results and Discussion

As previously mentioned, participants were asked to take photos of what they considered best captured their migration experiences, without offering any additional indications. Naturally, the photos taken covered a wide area of events, happenings, feelings, perceptions and impressions. Part of these referred also to the barriers and difficulties encountered in the migration process: mainly feelings of isolation, solitude or the language barrier. However, most of the photos shared by participants focused on the positive aspects associated with the migration journey. Consequently, we decided to center the discussion around the positive side of their experience, having in mind also the extensive existing literature on the negative aspects that migrants usually face in the host country.

### 4.1. Natural Beauty

One powerful idea that emerged from the Photovoice was the emotional refuge found in the natural beauty of Romania. All migrants who participated in the Photovoice shared at least one photo depicting a landscape or an image capturing the natural scenery. We selected three examples of such photos, shared by three different participants (Figures 1–3). The natural landscape is considered an important asset of Romania as a destination country, a characteristic stressed by many participants, as can be observed in the texts accompanying Figures 2 and 3.

The photos linked with the topic of nature are landscapes depicting the natural beauty of the scenery (Figure 2) or landscapes that also include the migrant herself (Figures 1 and 3). Including themselves in the pictures is a testimony of the migrants' feeling of belonging to the scenery and to the host country. Locating oneself in the natural setting of the new country is a key feature for the migrants' willingness to integrate in the host society and for their actual integration [22]. What is more, the author of Figure 1 revealed—during the group discussion following the second art-based session—that, from all her photographs submitted, this precise one was her favorite. These findings confirm previous evidences from the literature [22,23] arguing that placing one's self in nature-based settings is a key aspect to migrant integration and "*the ability to find points of connection with the landscape is central to integration into a new society*" ([22], p. 526).

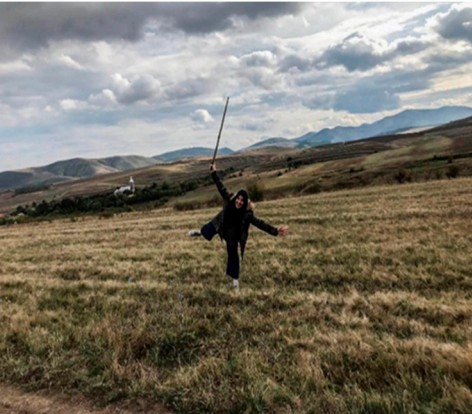

**Figure 1.** Photovoice picture illustrating the beauty of nature. "*I was really different from those around me, I just started looking for what I love from nature, discovering and experiencing that first feeling that I've been waiting for 21 years*" (23-year-old migrant from Palestine).

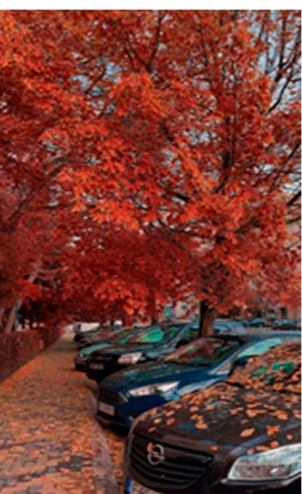

**Figure 2.** Photovoice picture illustrating the beauty of urban nature. *"The first thing that attracts a foreigner is the charming Romanian landscape: mountains, seaside, woods, parks, etc. Romania is a country where we can still breath fresh air, without any suffocating pollution. Those who love nature will be pampered here, just like me!"* (29 y.o., Algeria).

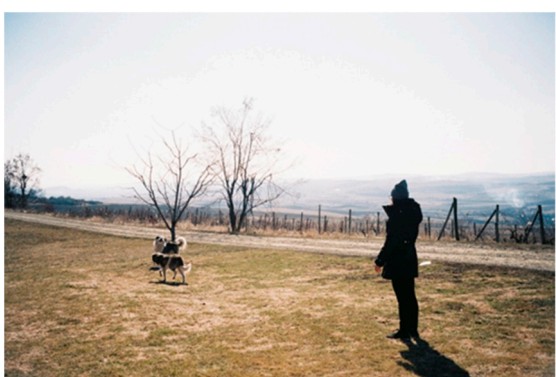

**Figure 3.** Photovoice picture illustrating the beauty of countryside nature. *"What I consider to be the strength of Romania: its fields and nature, in general. The photo was taken in the village of Batoș, in Transylvania, where you can find a lot of vineyards"* (20 y.o., the Republic of Moldova).

Being part of the natural setting is an essential experience in discovering the pleasant aspects of life in the new country [41]. One participant mentioned that, upon arrival in the new country, she started to look for some familiar natural settings, to discover and experience the surroundings, which offered her a feeling of engagement with the natural space (Figure 1). Moreover, by spending time in nature for leisure, migrants can benefit from positive emotions, such as feelings of peace and safety, which can reduce acculturation stress in the new country. Therefore, engaging in activities that take place in nature contributes not only to the development of a new identity in the receiving country, but also to better mental health. Charles Rodriguez et al. [41] offer an extensive review of the research highlighting the relationship between nature and migrants' integration. Their results are in line with the theme emerged from our Photovoice activity: nature and exposure to the natural setting enhances migrants' health and wellbeing and promotes their adaptation to the host country [25,41,42]. To reinforce the idea above, it is also worth mentioning that each participant submitted at least one photo with a natural landscape, confirming nature as one of the main advantages and opportunities offered by the host society and by the overall migration experience [19,20]. Thus, the Photovoice findings support the well-established perception from the literature that connecting to nature leads to strengthened attachments to the host society and to facilitated integration patterns [41,43].

*4.2. Education*

Generally, the literature shows that the main reasons for migration are related to labor opportunities, economic incentives, living conditions, and an overall better life quality, while education plays only a secondary role. Consequently, education is not viewed as a main push or pull factor for migration, except for the case of young individuals who decide to continue their studies abroad [44]. Since the participants in our Photovoice project were only young migrants, it was thus expected for education to emerge as one of the opportunities associated with the migration experience. As such, Figure 4 refers to the educational opportunities offered by migration. The participant who took this photograph submitted two pictures having the same topic, both entitled "The powerful weapon", one of which is presented in Figure 4. The photograph represents the Central University Library in Bucharest, a strong symbol of education in Romania. The text accompanying the picture stresses the importance that education has for the author and for her decision to migrate. Although also available in the home country, accessibility to education there might not be straightforward; hence, seeking alternatives in other countries turns education into a primary trigger of migration. Since there is a deliberate decision to study in the host country, education becomes one of the main drivers of migration, and not just a secondary cause among other socio-economic factors that could improve the quality of life [44]. The participant is thankful for the chance she received in the host country to follow her dream degree, which will open the way to the dream job and career. That is why she regards education in the destination country as a "powerful weapon", a competitive advantage she acquires.

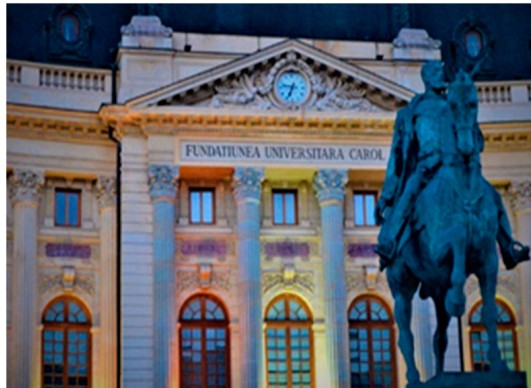

**Figure 4.** Photovoice picture illustrating Education—"The powerful weapon". *"I photographed the Central University Library to show the things I like most about Romania, Bucharest. The two reasons people migrate are better job opportunities and education. What I like here the most and what I am thankful for is the access to education. Although in Lebanon we have some of the best universities in the world and access to education is possible, pursuing it is not easy. This is highly due to political issues. In Bucharest, I had access to education and pursued my dream degree. Here, I acquired the powerful weapon"* (22 y.o., Lebanon).

Economic expectations are strongly linked with education, since a higher education would lead to better qualifications and in the end to a better job. This is also highlighted in the text accompanying Figure 4. Education is a chance for migrants to improve their economic condition and, for some of them, even the path out of poverty. Education and acquisition of skills are important aspects in many phases of migration. One of the main reasons to migrate is represented by the differential returns to skills in the home country compared to the destination country. There is an important string of literature showing that education serves as a catalyst for migration, especially for ambitious, middle-class young people, as highlighted by Browne [44]. Since we can consider our Photovoice participants

to belong to this category, the findings confirm their aspirations for higher education and better jobs. Although it is difficult to disentangle education and economic opportunities, the acquisition of education may be a stronger motivator for migration than higher wages, a theory sustained also by the increasing student migration flows [45].

Education also emerges in our Photovoice project as an important opportunity associated with the migration process through its role of key factor in the integration process. Education has a significant impact on both the probability of employment and on the level of wages, two important pillars in the economic and social integration of migrants [24]. Although not expressed directly by the participant, it is well understood that education, with its effects and implications, is also a "powerful weapon" for socio-economic integration.

### 4.3. Community, Society and People

For many migrants, the process of integration involves not only the adaptation to a new physical environment, but also to a new social and cultural context. As a consequence, a central aspect of the Photovoice evoked by the participants was the concept of community. Generally, a community is referred to as a heterogenous group of people, encompassing a range of attributes and linked by mutual affiliations, such as social bonds and shared outlooks, who collaborate on initiatives and inhabit a given geographic location or environment; joint efforts are a key feature of community life and individuals within the group work together to achieve shared goals and objectives [46]. Such collaborations typically reflect the culture and customs of the community, resulting in a sense of shared identity and purpose.

The concept of community might have a broad definition, so migrants will most probably also perceive it differently depending on the quantity and quality of their interactions with members of the host society. MacQueen et al. [46] stated that besides sharing a common interest, community also means joint actions and multiculturalism.

Figure 5 shows a person with a henna design on her hand sitting across another person with her hand outstretched. The person with the henna design is holding a small cone filled with henna paste and is carefully applying it onto the outstretched hand of the other person, creating a tattoo. This picture captures two aspects: the sense of cultural exchange and connection, and the act of sharing the henna application process, which stands for the sense of community and belonging. Henna application is a traditional practice that is part of celebrations and social gatherings in some cultures, as Basas [47] observed. Therefore, this activity is a way of sharing traditions and breaking down the boundaries between two different cultures.

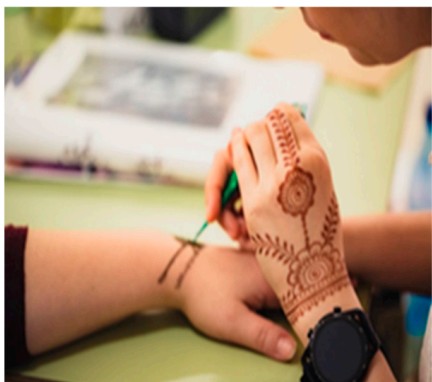

**Figure 5.** Photovoice picture illustrating the feeling of community and the cultural displacement. "*I arrived in Sibiu as a result of the war in Syria. Soon after, I enrolled in college and got to know the city and its people. I immediately realized that most of my Romanian colleagues and friends understood the Arab world in terms of bombs, shawrma, desert and camels. By establishing an Arab Cultural Center, I proposed to make at least some of these aspects of the culture and civilization of the Arab peoples known in Romania*" (32 y.o., Syria).

At the same time, the author of Figure 5 mentioned that upon her arrival in Romania, she observed that the general opinion about Arabs was shaped by war conflicts and culinary habits. So, this observation motivated her to establish an Arab Cultural Center, an organization through which she was not only able to keep her traditions alive, but also to enhance the local population's understanding and awareness of Arabic culture, language, and religion. The Center helped dispel negative stereotypes associated with Arabic culture and contributed to a change in the mentalities of the local population concerning said culture. Working at the Center allowed her to develop new managerial skills. Still, more importantly, it offered her a feeling of fulfilment by contributing to the better integration of other Arab migrants in Romania. Additionally, the Center helped the local population become more familiar with Arabic culture, working towards creating a more inclusive society that embraces diversity.

The Romanian community was described by the participants as inclusive, supportive, and accepting of cultural diversity. Figure 6 depicts two girls with smiles on their faces, who are embracing each other with their hands raised up in the air. The image conveys a sense of happiness, friendship, and celebration, suggesting that the girls are relaxed and comfortable in each other's company. The picture shows the power of friendship and community in bringing people together and creating moments of happiness and celebration. The author of this photograph expressed positive feelings towards Romanians, considering them as open-minded and friendly, making it easy to meet new people and form connections. Also, she mentioned that the strong sense of safety in the country encourages socialization and going out, as people can feel secure walking around even late at night without worrying about their personal safety. The feeling of safeness as a key to faster integration was also mentioned by authors such as Stewart and Mulvey [48] or Kearns and Whitley [49]. This sets Romania apart from other countries where safety concerns may hinder social activities.

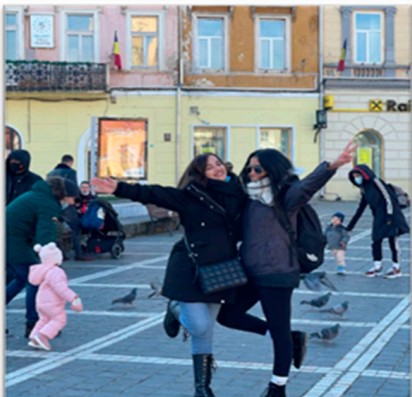

**Figure 6.** Photovoice picture illustrating the warm and hospitable nature of Romanians. *"The Romanian population is very warm, very welcoming, always available to help anyone. They are very open and friendly. (...) Also, security is the main point which encourages meetings, going outs, etc. Sometimes, I go for a walk with friends and return at 2 a.m. without ever being approached by anyone, something that is rare in other countries"* (29 y.o., Algeria).

The Photovoice findings highlight the positive experiences that migrants have in their new social context, such as moments of connection and support from the local population [50,51]. These results confirm that societies play an important role in the integration process and help to counter vulnerabilities of migrants [50,52,53].

*4.4. Architecture and Cultural Experiences*

Architecture emerged through the photos of our participants as a way of expressing feelings toward their new environment, as well as a proxy for emphasizing the cultural

diversity in Romania. Architecture, as a powerful form of cultural expression, reflects the history, traditions, and values of a place or community [52]. According to Mosley et al. [53], architecture is not limited to conscious design choices, but can also have unconscious effects; buildings and spaces can have a profound impact on the emotions and behavior of people, and on the ways in which they perceive the surrounding world, i.e., the host society. The Photovoice participants opted to showcase images that featured architecture, underscoring the diverse cultural makeup of the Romanian community. Their choices highlight the unique blend of influences that have shaped communitarian identity, reinforcing the idea that Romanian culture is a compound of different traditions and artistic styles. The assortment of images, from old to new and through different styles, calls attention to the rich architectural history of this region and its cultural diversity, while also providing a unique perspective on the participants' experiences in their new environment. The participants were able to showcase the importance of cultural exchange and understanding and how architecture can serve as a powerful visual representation of a community's heritage and values.

Figure 7 depicts the panoramic view of water fountains at dusk taken from a balcony. The sky above is a stunning gradient of warm oranges, yellows, purples, and ultramarine, indicating that the sun has just set below the horizon. The light from lamps gives the fountains a soft and magical glow, creating a serene feeling of tranquility. The author has effectively conveyed the ambiance of the picture through the narrative and also mentioned the soothing sound of the water, which blended seamlessly with the ambient sounds of nature.

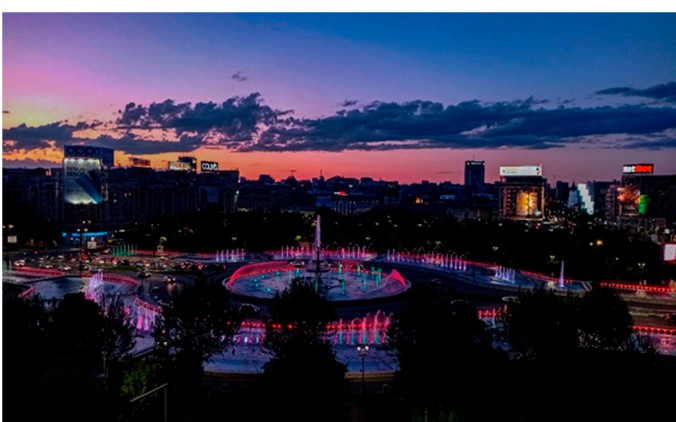

**Figure 7.** Photovoice picture illustrating the panoramic view of water fountains at dusk. "*We were at the end of the summer. We were invited to a meal with a relative during Ramadan. It is our Arab custom to respect the council, i.e., the obligation to sit with everyone, especially if it was your first visit. I went to the kitchen to sip some water, I knew at that time that they have a balcony overlooking this beauty, so I entered the balcony shyly, and I felt the first cold sip that warmed my heart. The sound of the water was like a symphony with the sound of the wind. Its colours are in harmony with the colours of the enchanting twilight, I am not exaggerating when I describe it as a painting*" (23 y.o., Palestine).

In Transylvanian cities, the Baroque can be observed on and in many buildings. Figure 8 depicts a historic edifice located at the heart of Brasov, boasting Baroque characteristics that are most prominently reflected in its ornate carvings, intricate sculptures, richly decorated moldings, and grand archways. This picture emphasizes how migrants encounter cultural differences between their country of origin and the host country. Experiencing these differences can be both overwhelming and exciting for migrants, as they have to learn how to navigate through new cultures and adapt to a new way of living.

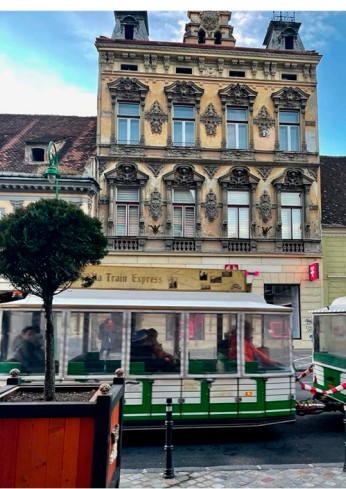

**Figure 8.** Photovoice picture illustrating old architecture. *"All architectural styles interfere in Romania. From vernacular to modern. While strolling in the streets one can easily travel in time through the architectural varieties of the buildings. How not to enjoy this beautiful country once you have learned the language!"* (29 y.o., Algeria).

## 5. Conclusions

Participatory Action Research was used in this study to investigate the perceptions and realities of migrants in their destination country through an immersive artistic experience. This approach aimed to gain insights into these individuals' transformational journeys [38]. The results obtained from the analysis are filling the gap in the literature on the process of migrants' integration in Romania, a country usually associated with net emigration; the significant increase in migration to Romania over the last years is worth mentioning [16]. Migrants were invited to reflect on their integration experiences and provide honest feedback, with results being free of any academic prejudices or biases. The eight participants from the Republic of Moldova or Arabic-speaking countries adopted a self-selection approach based on their familiarity with arts and visual performance. The results were analyzed using ethnographic methods, which included studying photos and their descriptive texts along with notes from three participatory sessions. The aim of this approach was to gain a deep understanding of the participants' experiences and perspectives. Considering the small number of participants, comparing the experiences in terms of ethnicity was not the goal of the analysis. The data was carefully analyzed to identify patterns and themes, which helped to interpret the findings. Several topics emerged, related to nature, education, community and people, culture, and architecture. To extrapolate the results, further quantitative researches are needed. Overall, this approach allowed for a comprehensive and nuanced analysis of the data, resulting in meaningful insights into the research at hand.

Natural beauty is considered a valuable asset for Romania, the theme being highlighted in several pictures. Nature was found to be an emotional refuge, as well as a bridge between the former and current environment. Moreover, immersion or integration in natural settings expresses the willingness of participants to be part of the host society, as well as an essential experience in belonging to the host society.

Education emerged as one central theme related to the migration experience. This should not be a surprise, as all participants were young. Access to education in the home country may not have been easy, despite its availability. There could have been several barriers, including financial constraints, lack of proper infrastructure, or social taboos. For instance, Arab girls may not have been allowed to attend school due to gender stereotypes prevailing in the society, while children from low-income families may have had to forego education to contribute to the family's income. Thus, education was represented as a "powerful weapon", a chance to improve the economic condition of migrants, a path out of poverty, but also a key factor in their integration process.

Integrating into a new country involves more than just adjusting to a different culture; it also requires sharing one's own traditions and promoting cultural exchange to bridge the gap between two distinct societies. In most of the cases, migrants participated in the Photovoice emphasized that Romanians usually stand out for being open-minded, supportive, and friendly towards migrants, allowing them to engage freely in social activities. The participants found integration into local communities an easy process, these findings being in accordance with previous studies [11,16]. Finally, the participants proved through the photos they submitted that architecture can be a powerful visual representation of a community's values and traditions.

The Photovoice project gave migrants a platform to share their personal experiences and feel empowered in their integration journey. Through this opportunity, they were able to inspire their peers and create a more authentic dialogue about the challenges and joys of adapting to a new destination. This exploratory study demonstrates the capacity of PAR to portray genuine migration experiences of young individuals. However, we are aware that the sample selection is crucial, and a self-selection approach could lead to sample-specific results. Nonetheless, PAR provides valuable insights that could complement existing knowledge on the specific group of migrants in Romania.

Education and communities play a strong and positive role in promoting integration, according to existing research. Additionally, our study found that nature and cultural adventures can also act as facilitators for integration and positive migration outcomes. This novel finding highlights the potential for incorporating outdoor and cultural activities into integration programs and offers new insights into methods for supporting migrant populations. Natural beauty and architecture were not generally portrayed as integration facilitators in previous research on migration into Romania [11]. The Photovoice method has enabled participants to discover new integration factors, including nature and architecture, and to reflect on their previously unexplored benefits and relevance. Through this process of closely examining their surroundings and capturing them in photographs, participants have uncovered unexpected insights and perspectives.

**Author Contributions:** Introduction, M.R., V.I.R. and S.C.; background: M.R., E.-M.P., S.C. and I.M.; materials and methods, E.-M.P.; data collection, all; validation, M.R.; formal analysis, E.-M.P., S.C. and I.M.; investigation, M.R., E.-M.P., S.C., I.M. and V.I.R.; review and editing, M.R. and V.I.R.; visualization, M.R.; concept and supervision, M.R.; project administration, M.R. All authors have read and agreed to the published version of the manuscript.

**Funding:** This research was funded through the European Union's Horizon 2020 research and innovation program under Grant Agreement No. 870700: "*Empowerment through Liquid Integration of Migrant Youth in Vulnerable Conditions (MIMY)*".

**Institutional Review Board Statement:** This research was conducted according to the ethical guidelines of the MIMY project leadership (ERP 19-055 MIMY; 6 March 2020).

**Informed Consent Statement:** Informed consent was obtained from all subjects involved in the study.

**Data Availability Statement:** Not applicable.

**Conflicts of Interest:** The authors declare no conflict of interest.

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
