# Peer review of "“A Picture Is Worth a Thousand Words”: Youth Migration Narratives in a Photovoice"

_societies, doi:10.3390/soc13090198_

Round 1

Reviewer 1 Report

Dear authors,

A very interesting approach to researching integration of immigrants. What is missing is a clear description of PAR and making clesar that photovoice is just one possible approach. Honestly, I think that the research is strongly action based and definitely art-based, but the participatory approach is a bit weak - it seems the participants could not decide if they want to show challenges to integration as well or use different methods (like video, sound recording) which would be essential in PAR. 

I suggest to make some relativisations to your presumptions that frame the whole paper: Your paper focus only "on the positive experiences of young vulnerable migrants" (p1) and you make some assumptions like "migrants arrive in a new country with aspirations and dreams for an improved life"... (which does not really hold for refugees, like recently from Ukraine)

Your description of the use of photovoice on different research topics is sometimes misleading - or your interpretation is very free, like regarding Miterko, P.; Bruna, S. 2022: the article is on homeless people, it is argueable if the results can be interpretetd 1 to 1 for migrants. 

The presentation of the literature you found after the photovoice excercise in a dedicated section seems not perfect. Did you try using more or less the same statements after your presentation of the images? Like showing the images of nature and afterwards supporting your interpretation with (a shortend) paragraph on engagement with nature as integration facilitator.

Concerning the literature you are using in your interpretation:

Migration and labour market is in your paper reduced to work life and safety, there is more literature on the role of employment for integration - but employment is not really coming up as a topic in the presented images.

Social capital and community - this part is weak, since there is so much research on social capital, its connection to cultural capital and the impact on integration that the chosen literature seems very random. Furthermore, it is questionable if the used description of "a community " following MacQueen can be transfered to Europe, where multiculturalism is not common.

Simlar holds for architecture and city development.

All in all the paper is very interesting but could be improved by

a) describing the method of PAR and reducing a bit to the disadvantage of the description of photovoice.

b) describing the sample (8 females, aged 19 to 32, 1 from Moldova, 1 from Syria...., 1 student visa, 2 refugees, 2 asylum seeker) since this gives a better framework for understanding the results.

c) focussing on the images and their interpretation (by presenting supporting literature after presenting the images) and including more results of the discussions

d) using more EUropean literature (e.g., Germany, France, NL, Sweden, Finland) providing a slightly different framework for "integration" than US, and which gives also the frame for integration in Romania.

a) and c) seem the most important ones for improvement

in the file you will find some comments and some highlighted passages which are unclear.

all the best

Some of your sentences are difficult to understand and maybe not correct. e.g.

"The latter ones were circumsribed into several topics" .... easier: "the images were grouped in different categories"

Reviewer 2 Report

It is clearly a well-written article based on relevant sources, methodological innovation and practical importance for the problem of migrant integration. Both theoretical considerations and empirical research seem to me to be sufficiently elaborated for a good academic piece.

I suggest publication. However, in my opinion, some improvements may be needed which focus mostly on style of argumentation, the level of analysis and the attention for the limitations of the study.

Perhaps the (sub-)title of the article could specify the country under consideration. Also, keywords could be reassessed, as, for instance, “opportunities” or “integration” are too vague and general to guide the potential reader towards the article.

The introduction assumes aspirations for an improved life to be “the” motivation for migration while research on motivation reveals a more complex picture (current boom of literature on migrant remittances stands out as an example). The same applies for the assumption that migrants desire integration which is not always the case. And also, migrants which are the object of the research are initially presented as “vulnerable”. This vulnerability should be explained. Perhaps these things could be specified in terms of the aims of the article.

The structure of the background section is a little unconvincing. Human and social capital are listed as key integration factors; nevertheless, they appear in the middle of the section, and moreover, their relation to the relevant topics of the research (natural environment, etc.) is not clarified enough. On a terminological note, the use of “integration factors”, “integration catalysts” and “integration facilitators” seems interchangeable.

I am not certain whether this research requires anonymity of participants in the study. If yes, then concrete references such as the Arab Cultural Center in Sibiu may be revealing.

The article is explicitly devoted to positive experiences of migrants. Accordingly, not a single negative experience is discussed. What is the reason for that? Isn’t it possible that the participants in the PAR perceive their task to share only positive experiences? Perhaps some short discussion on the expected outcomes of the research is needed.

Next, distance from the object of research is usually a difficult achievement. There are some examples in the manuscript in which the reader may be confused on whether it is about the viewpoints of the migrants or the conclusions of the researchers. Here is just one case: “Romania stands out for being open-minded, supportive and friendly towards migrants. The country is known for its safety, allowing migrants to engage freely in social activities. Romanians are welcoming towards foreigners, making it easier for them to integrate into their communities.” It rather sounds like a touristic brochure advertising Romania.

The sample is composed of representatives of two rather different ethnic communities (from Moldova and from the Arab countries). Having in mind the small sample, it is difficult to compare migrant experiences in terms of ethnicity. Nevertheless, adding some notes on that could be useful.

Finally, the conclusion may benefit from elaborating on the research outcomes for future research in the context of what has already been established in the literature. For me, it seems interesting since Romania has been traditionally perceived as a migrant-sending rather than a migrant-receiving country. Therefore, migratory experiences in this particular situation could shed additional light on processes of integration in host societies.

Overall, I believe that these suggestions do not imply a substantial reorganization of the article and could be of use to the author(s) for better presentation of their results.

Round 2

Reviewer 1 Report

Thank you for addressing all issues mentioned in  the last review report.